# ssFPN: Scale Sequence (*S*^2^) Feature-Based Feature Pyramid Network for Object Detection

**DOI:** 10.3390/s23094432

**Published:** 2023-04-30

**Authors:** Hye-Jin Park, Ji-Woo Kang, Byung-Gyu Kim

**Affiliations:** Department of Artificial Intelligence Engineering, Sookmyung Women’s University, 100 Chungpa-ro 47 gil, Yongsna-gu, Seoul 04310, Republic of Korea

**Keywords:** object detection, feature pyramid network, scale sequence (*S*^2^) feature, convolutional neural network (CNN), deep learning

## Abstract

Object detection is a fundamental task in computer vision. Over the past several years, convolutional neural network (CNN)-based object detection models have significantly improved detection accuracyin terms of average precision (AP). Furthermore, feature pyramid networks (FPNs) are essential modules for object detection models to consider various object scales. However, the AP for small objects is lower than the AP for medium and large objects. It is difficult to recognize small objects because they do not have sufficient information, and information is lost in deeper CNN layers. This paper proposes a new FPN model named ssFPN (scale sequence (*S*${}^{2}$) feature-based feature pyramid network) to detect multi-scale objects, especially small objects. We propose a new scale sequence (*S*${}^{2}$) feature that is extracted by 3D convolution on the level of the FPN. It is defined and extracted from the FPN to strengthen the information on small objects based on scale-space theory. Motivated by this theory, the FPN is regarded as a scale space and extracts a scale sequence (*S*${}^{2}$) feature by three-dimensional convolution on the level axis of the FPN. The defined feature is basically scale-invariant and is built on a high-resolution pyramid feature map for small objects. Additionally, the deigned *S*${}^{2}$ feature can be extended to most object detection models based on FPNs. We also designed a feature-level super-resolution approach to show the efficiency of the scale sequence (*S*${}^{2}$) feature. We verified that the scale sequence (*S*${}^{2}$) feature could improve the classification accuracy for low-resolution images by training a feature-level super-resolution model. To demonstrate the effect of the scale sequence (*S*${}^{2}$) feature, experiments on the scale sequence (*S*${}^{2}$) feature built-in object detection approach including both one-stage and two-stage models were conducted on the MS COCO dataset. For the two-stage object detection models Faster R-CNN and Mask R-CNN with the *S*${}^{2}$ feature, AP improvements of up to 1.6% and 1.4%, respectively, were achieved. Additionally, the AP${}_{S}$ of each model was improved by 1.2% and 1.1%, respectively. Furthermore, the one-stage object detection models in the YOLO series were improved. For YOLOv4-P5, YOLOv4-P6, YOLOR-P6, YOLOR-W6, and YOLOR-D6 with the *S*${}^{2}$ feature, 0.9%, 0.5%, 0.5%, 0.1%, and 0.1% AP improvements were observed. For small object detection, the AP${}_{S}$ increased by 1.1%, 1.1%, 0.9%, 0.4%, and 0.1%, respectively. Experiments using the feature-level super-resolution approach with the proposed scale sequence (*S*${}^{2}$) feature were conducted on the CIFAR-100 dataset. By training the feature-level super-resolution model, we verified that ResNet-101 with the *S*${}^{2}$ feature trained on LR images achieved a 55.2% classification accuracy, which was 1.6% higher than for ResNet-101 trained on HR images.

## 1. Introduction

Deep learning models have shown substantial progress in various computer vision tasks and their applications [1,2,3,4,5,6,7,8,9]. Object detection is a fundamental computer vision task that can be applied in object tracking [10,11], segmentation [12], pose estimation [13], 3D object detection [14], autonomous driving [15], and unmanned aerial vehicle (UAV) systems [16]. Recently, convolutional neural network (CNN)-based object detection models have achieved remarkably improved performance in terms of average precision (AP) [17]. The initial object detection models comprised two detection steps: a classification step and a bounding box regression step. Before final prediction, the bounding box proposals were generated in the region proposal network (RPN) [18]. To improve the inference speed, one-stage object detection models such as YOLO were proposed [19]. The one-stage detectors integrated two steps, and therefore these object detection models could be operated in real time.

However, small object detection is still a very challenging task [20]. According to the MS COCO definition, an object is classified as “small” if the area of the segmentation mask is less than 32 × 32 pixels [21]. Additionally, if the area is greater than 32 × 32 pixels and less than 96 × 96 pixels, it is classified as “medium”. If the area exceeds 96 × 96 pixels, it is classified as “large”. Small object detection is challenging for several reasons. First, small objects have little information to be represented. It is difficult to separate objects from the background and discriminate similar categories. In addition, small objects can be located at various points in the image. It is hard to localize bounding boxes. Finally, most object detection models have been studied for large objects [22].

Recently, state-of-the-art models have been reported for detecting small objects. Usually, the average precision for small objects (AP${}_{S}$) is lower than the AP for medium (AP${}_{M}$) and large objects (AP${}_{L}$). Figure 1 shows the proportion of different object scales and the performance gap in the AP between small-, medium-, and large-scale objects in the MS COCO validation dataset. Small objects made up the largest proportion of the dataset. However, the average precision for small objects (AP${}_{S}$) was the lowest among the scales. The red line shows the performance gap compared to the other scales. For Cascade R-CNN, which is a two-stage detector, AP${}_{L}$ had the highest value at 54.4%, but AP${}_{S}$ had the lowest value at 24% [23]. The performance gap between AP${}_{L}$ and AP${}_{S}$ was 30%. Likewise, YOLOR-D6, which is a state-of-the-art architecture, had the lowest AP for small objects among the scales [24]. For YOLOR-D6, there was a 28% gap between AP${}_{S}$ (40.4%) and AP${}_{L}$ (68.8%). Small objects made up the largest proportion (41%) of the MS COCO dataset. Thus, if the performance of small object detection can be improved, the overall performance will naturally improve.

As objects have various scales in natural images, object detection models have to learn multi-scale features. To deal with multi-scale features, scale-invariant features have been studied in traditional computer vision [25]. A scale-invariant feature is detectable even if the object scale changes. If a model learns scale-invariant features, the small object detection problem can be solved efficiently. Scale space, which is a multi-scale representation, is parameterized by the variance of the Gaussian kernel to extract scale-invariant features [26]. Multi-scale representation can be composed of images with different resolutions.

On the other hand, recent deep-learning-based object detection models have used feature pyramid networks (FPNs) [27] as a neck module to handle multi-scale objects effectively. Before detecting objects in the head, they are assigned to a single pyramid level according to their scale. For example, large objects are detected in a low-resolution pyramid feature map, whereas small objects can be detected in a high-resolution pyramid feature map. To improve the performance, FPN-based models have been proposed to alleviate the semantic gap between each level of the pyramid feature map [28]. However, most of the existing models use simple fusion operations such as concatenation. Therefore, they cannot sufficiently consider the correlation of all pyramid feature maps.

When an input image is fed into the CNN, output feature maps of each convolution layer are composed by the FPN. The resolution of the pyramid feature maps decreases as they pass through the convolution process. This FPN architecture is similar to scale space, and the level axis of the FPN can be considered as the scale axis. Therefore, scale-invariant features can be extracted from the FPN as in [29]. This approach motivated us to propose a scale sequence (*S*${}^{2}$) feature for the FPN. The higher the pyramid level, the smaller the image size. However, the semantic information is enhanced. We considered the level axis of the FPN as the time axis of the sequence and extracted spatio-temporal features by 3D convolution [30]. As a result, the scale sequence (*S*${}^{2}$) feature could be a unique feature of scale space, and it is scale-invariant. Furthermore, all of the FPN feature maps could participate in the operation using 3D convolution. This model includes a scale correlation between all pyramid feature maps.

In comparison to other scales, the reason for the small object problem is that the deeper CNN layer leads to a loss of information such as small object features and localization information for bounding boxes [22]. For small objects, we designed a scale sequence (*S*${}^{2}$) feature built on a high-resolution pyramid feature map. Generally, small objects are detected in high-resolution pyramid feature maps. Therefore, we resized each pyramid feature equally to form a high-resolution feature map. Pyramid feature maps with extended resolution are similar to Gaussian pyramids. They are concatenated to a 4D tensor for 3D convolution. This cube feature can be considered as a general view of the dynamic head presented in [31]. The designed scale sequence (*S*${}^{2}$) feature is concatenated to a high-resolution pyramid feature map for detecting small objects in the head.

The contributions of this paper are four-fold:We proposed a new scale sequence (*S*${}^{2}$) feature that is extracted by 3D convolution on the level of the FPN. It is a scale-invariant feature of the FPN regarded as scale space.The scale sequence (*S*${}^{2}$) feature could improve the AP for small objects as well as the AP for other scales, since it was built on a high-resolution feature map to strengthen the features of small objects.The scale sequence (*S*${}^{2}$) feature could be extended to most object detection models based on the FPN structure. We verified the performance of the scale sequence (*S*${}^{2}$) feature in one-stage and two-stage detectors.The scale sequence (*S*${}^{2}$) feature could improve classification accuracy on low-resolution images by training feature-level super-resolution models.

## 2. Related Works

### 2.1. Object Detection

Small object detection has been continuously studied in the field of computer vision. The most commonly used approach is a featurized image pyramid strategy. This architecture uses multi-scale image resolution as the input and predicted objects in output feature maps of each model. It allows the model to learn images of various scales.

For example, Azimi et al. jointly employed image cascade networks (ICNs) and feature pyramid networks (FPNs) to detect small objects in remote sensing images [32]. This design leveraged the feature pyramid hierarchy to extract strong semantic features of different scales. Likewise, the high-resolution detection network (HRDNet) used images with various input resolutions [33]. In addition, this model utilized different backbone models for input resolution. For example, a small input image ($I_{2}$) was fed into the deeper CNN network; in contrast, a large image ($I_{0}$) was fed into a shallow backbone network. Finally, the output FPNs of each backbone were fused. This architecture achieved improvements for small object datasets such as VisDrone2019.

In addition, Chen et al. proposed an architecture for a scale attention model that utilized two different scales of images [34]. This study applied the attention mechanism to learn the scale. The scale attention model was trained for large objects using images with a scale of 1. On the other hand, the model learned small and tiny objects from images with a scale of 0.5. Two CNN networks were jointly trained to learn the scales, and the score maps were fused for prediction. Tao et al. proposed hierarchical multi-scale attention for multi-scale inference [35]. They utilized an attention mechanism from previous research [34] but reconstructed it hierarchically. This model predicted the attention value of adjacent scale pairs when training different scales. Then, the model hierarchically combined multiple scale prediction values at the inference time.

The abovementioned methods improved the performance of small and tiny object detection. However, they are not tractable in terms of memory and computation in real time, because multi-scale input images are very challenging in computer vision. In this paper, the proposed ssFPN used only a single image as input to reduce complexity and improved the performance of small object detection.

### 2.2. Feature Fusion Strategy

Feature pyramid networks (FPNs) are essential modules for handling multi-scale features [27]. FPNs comprise different resolutions of feature pyramids, to which objects are assigned according to their scale. These feature pyramids are fused by a top-down pathway. However, there is a discrepancy problem between each pyramid feature map, because they are generated from different convolution layer depths. Additionally, low-level features are reflected to a smaller extent in the final prediction.

Path aggregation networks (PANets) have been proposed for a new fusion method to alleviate the problem by adding a bottom-up pathway to the FPN [28]. PANets using both high-level and low-level features have been proposed. PANets utilize bottom-up path augmentation, which connects low-level features to high-level features, and then a new feature pyramid is created (denoted as N). This leads to accurate bounding box localization prediction due to plentiful low-level features. Finally, the generated vectors in each feature map are combined using adaptive feature pooling. The PANet is an effective feature fusion model, and it was selected for the neck module of YOLOv4. NAS-FPN found an effective feature pyramid architecture by AutoML [36]. NAS stands for neural architecture search. The previous methods were designed by handcrafted fusion connection. However, NAS-FPN searched for the best feature fusion method by a reinforcement learning scheme. Additionally, a bidirectional feature pyramid network (BiFPN) pointed out other models considering all pyramid feature maps equally regardless of their resolution [37].

Tan et al. proposed a weighted fusion method for feature pyramids [37]. Liu et al. proposed a composite backbone network architecture (CBNet) [38]. CBNet used dual or triple backbones to strengthen features for training. There were two types of backbone: the lead backbone, which connected the detection head, and the other assistant backbones. Then, CBNet fused the output feature maps from each backbone using the adjacent higher-level composition (AHLC) method. CBNetV2 applied a training strategy with auxiliary supervision [39]. Since multiple backbones were used, additional optimization was required. CBNetV2 used two detection heads. One detection head was trained by the original detection loss, and the other detection head supervised assistant backbones to optimize the learning process.

Recently, the dynamic head introduced three attention modules: scale-aware attention, spatial-aware attention, and task-aware attention [31]. The scale-aware attention module was trained according to the importance of adapting the pyramid level to the object scale. Additionally, the dynamic head utilized a general view that concatenated the pyramid feature maps after resizing them to medium feature map resolution. Finally, the self-attention between the feature maps of the general view was computed.

However, most previous studies have fused pyramid features by simple summing and concatenation. This simple structure cannot sufficiently consider the correlation between all pyramid feature maps. In this paper, the proposed scale sequence (*S*${}^{2}$) feature was concatenated to pyramid feature maps. It reflected the correlation across the whole feature pyramid by participating in 3D convolution. Therefore, it could enrich the feature information of the FPN for detecting multi-scale objects.

### 2.3. Scale-Invariant Features

A scale-invariant feature is defined as a feature that does not change even when the object scale changes [25]. In traditional computer vision, scale-invariant features have been studied to deal with multi-scale objects. Image pyramids, which are a basic approach, can represent various scales of objects.

Scale-invariant feature transform (SIFT) extracts features that are robust to scale and rotation. These scale-invariant features can be found through scale-space theory [40]. First, an image pyramid is created by resizing an original image. To create a scale space, Guassian filters with different scale factors are applied to the image pyramid. Then, extrema can be detected by the difference of Guassians (DoG). These extrema become candidates for key points, and the most stable key points are selected as scale-invariant features. The SIFT algorithm has traditionally been widely used for image matching tasks.

Meanwhile, some studies have considered scale correlation in feature pyramids instead of image pyramids to reduce computation complexity. The deep-scale relationship network (DSRN) was proposed to address large variations in text scale [41]. DSRN utilized two modules: a scale-transfer module and a scale-relationship module. The scale-transfer module unified the dimensions of the feature maps for aggregation. The scale-relationship module aggregated multi-scale features from different stages and mapped them onto scale-invariant space. This model effectively handled the scale problem. DSRN improved performance, demonstrating a fast inference speed on a scene text detection dataset. Additionally, pyramid convolution (PConv) considered feature pyramids as scale space and extracted scale-invariant features [29]. Three convolution kernels were designed for three adjacent sizes of feature maps. These were used to extract the scale and spatial features between them. The convolution kernel size was the same, but the stride value was different. After convolution, three output feature maps were integrated. Furthermore, a scale-equalizing pyramid convolution (SEPC) method using deformable convolution was proposed. PConv improved the performance in terms of AP on the MS COCO dataset.

However, these approaches computed the convolution of each pyramid feature map independently. In this paper, the FPN was regarded as a scale space, and we extracted scale-invariant features by 3D convolution. The extracted features were defined as scale sequence (*S*${}^{2}$) features that were unique features of the FPN. All pyramid feature maps were computed by 3D convolution. Through this process, the correlation across all pyramid features could be considered sufficiently. Furthermore, the proposed scale sequence (*S*${}^{2}$) feature included sequence information of scale transformation.

### 2.4. Super Resolution

Super resolution is the task of generating a high-resolution (HR) image from a low-resolution (LR) image. It can be divided into image-level super resolution and feature-level super resolution, depending on the purpose of the application.

Image-level super resolution predicts the whole high-resolution image. In general, deep-learning-based super-resolution models involve image-level super resolution, which has been studied over the past several years [42,43,44,45]. An enhanced deep super-resolution network (EDSR) [45] was trained by residual learning, and many super-resolution models have been studied based on EDSR [45]. Lim et al. divided a deeper network into residual blocks for stable training and connected them using a skip connection. Furthermore, the EDSR generated high-resolution images through pixel shuffling. Some studies on the super-resolution approach have also been conducted in the field of object detection [46,47]. However, image-level super resolution is inefficient for object detection and classification tasks, which do not require the generation of the whole image. Additionally, it increases the complexity of models and can affect the inference speed adversely.

Recently, some feature-level super-resolution methods have been studied to reduce computational complexity [48,49]. These methods super-resolve the feature map, not the low-resolution image. Unlike image-level super-resolution, object detection or classification predicts targets based on semantic features from the last layer of the model. Therefore, it is not necessary to predict the whole image, which would bring a computational burden. Feature-level super resolution is sufficient to create high-quality super-resolved features for performance improvement when using low-resolution images. Noh et al. [49] proposed a super-resolution feature generator based on generative adversarial networks (GANs) [50].

Motivated by their approach, we introduce herein a feature-level super-resolution method using scale sequence (*S*${}^{2}$) features. Since the scale sequence (*S*${}^{2}$) features include the scale sequential information of the input images, they can help to generate super-resolved high-resolution features.

## 3. Proposed Architecture

### 3.1. Scale Sequence (S${}^{2}$) Features

This section introduces a new feature: the scale sequence (*S*${}^{2}$) feature. We aimed to find a scale-invariant feature of the FPN. A scale-invariant feature does not change when the size of the image is changed. First, this section explains scale-space theory [26] in traditional computer vision. Scale space is constructed along the scale axis of an image. It represents not one scale, but various scale ranges that an object can have. The space is generated by blurring an image using a Guassian filter instead of resizing the image directly.

Figure 2 shows a comparison of image sequences on different axes. Figure 2a represents the scale space using a Guassian filter on the scale axis, and Figure 2b presents a general view concatenated with identical resolution features on the level axis [31]. Figure 2c shows video frames on the time axis.

As illustrated in Figure 2a, the larger the scale parameter value, the more blurred the generated image. According to this theory, the scale indicates the level of detail of the image. In other words, a blurred image loses detail. However, the structural features of the image are prominent. It can be computed as follows:(1)$$g_{\sigma }\left(x,y\right)=\frac{1}{2\pi \sigma^{2}}e^{-\left(x^{2}+y^{2}\right)/2\sigma^{2}},$$
(2)$$f_{\sigma }\left(x,y\right)=g_{\sigma }\left(x,y\right)*f\left(x,y\right),$$
where f(x,y) is a 2D image; $f_{\sigma }\left(x,y\right)$ is generated by smoothing through a series of convolutions with a 2D Guassian filter $g_{\sigma }\left(x,y\right)$; and σ is a scale parameter as the standard deviation of the 2D Guassian filter, which is used in convolution. As a result, these images have the same resolution but different scale parameter values.

This paper considers the feature pyramid network (FPN) as a scale space. The FPN is composed of output feature maps through convolution layers when an input image is fed into the CNN. A low-level pyramid feature map is high-resolution and contains information on localization, especially for small objects. On the other hand, a high-level pyramid feature map is low-resolution, but it contains plenty of semantic features. These characteristics are similar to those of scale space, which features trade-off information on the scale axis. Based on this structure, the general view from the dynamic head [31] was concatenated with all pyramid features after resizing them to the same resolution. This is illustrated in Figure 2b, which shows that the feature representations differed according to the level axis. Finally, a unique feature of this general view was extracted from the scale view of the FPN.
(3)$$G={\left\{P_{i}\right\}}_{i=3}^{L},$$
where $P_{i}$ is a pyramid feature map from the i−th levels; the highest-resolution feature pyramid is $P_{3}$; and the general view G is generated by concatenating identical-resolution feature maps after resizing the pyramid feature maps to a specific resolution. The general view was constructed as a 4D tensor: G=(level·width·height·channel).

A unique feature of the FPN has to consider all general view feature maps. This paper was motivated by 3D convolution [30] in video recognition tasks. In this area, 3D convolution is used to extract motion in videos. Figure 2c shows video frames on the time axis. Motion involves the sequential as well as spatial information of frames. The pyramid feature maps of the general view are regarded as video frames; hence, the general view is a sequence of convolutions. The time axis of video frames can be considered the level axis of the general view.

A unique feature of the general view is defined as a scale sequence (*S*${}^{2}$) feature. It is extracted by 3D convolution on the level axis of the general view. The proposed scale sequence (*S*${}^{2}$) feature is a spatio-temporal feature of the general view, such as motion. Furthermore, all pyramid feature maps of the FPN contribute to the 3D convolution operation.

As a result, this approach includes scale correlation across feature pyramids. It differs from other FPN-based feature fusion methods that employ simple summing and concatenation between pyramid feature maps. The scale sequence (*S*${}^{2}$) feature is defined as follows:(4)$$S^{2}feature=\Theta_{s^{2}}\left(G\right),$$
where $\Theta_{s^{2}}$ is the scale sequence module based on 3D convolution. This module can extract the scale sequence (*S*${}^{2}$) feature from the general view. To apply 3D convolution, we regarded the level axis of the general view as the time axis of video frames; G=(time·width·height·channel). As time refers to the length of frames, it could be denoted as the number of levels of the general view.

### 3.2. Framework Based on Scale Sequence (S${}^{2}$) Module

In this section, $\Theta_{s^{2}}$, i.e., the scale sequence (*S*${}^{2}$) module, is explained in detail. Figure 3 shows the proposed scale sequence (*S*${}^{2}$) module framework, and Figure 4 illustrates the architecture in detail. Figure 3a depicts the neck module of the FPN for feature fusion, and Figure 3b shows the process of the proposed scale sequence module using 3D convolution. Additionally, Figure 3c,d depict the one-stage detector head and two-stage detector head, respectively.

Generally, an object detection model is composed of a backbone network, a neck module for feature fusion, and a detection head. The input image is fed into the backbone network. A CNN or transformer [52] are employed as the backbone to extract features. Convolution features obtained through convolution layers are denoted as {$C_{1}$, $C_{2}$, $C_{3}$, $C_{4}$, $C_{5}$}. This module selects only *C*_3_ to *C*_5_ among the output feature maps of the corresponding convolution layers, because low-level feature maps lack semantic information.

Next, the generated convolution features are aggregated by top-down and bottom-up fusion in the neck. The path aggregation network (PAN) architecture [28] is adopted instead of the FPN for effective multi-scale feature fusion. The pyramid features are denoted as {$P_{3}$,$P_{4}$,$P_{5}$}. Figure 3 illustrates a basic YOLOv4-P5 model with the proposed scale sequence (*S*${}^{2}$) module that employs an input size of 896 × 896. Therefore, the baseline model produces up to $P_{5}$. If the model uses an input resolution larger than 896 × 896, the scale sequence module can extend the maximum pyramid level up to $P_{6}$ and $P_{7}$.

Figure 3b shows the pyramid features fed into the scale sequence module. In this module, a general view is generated by concatenating all pyramid feature maps after resizing them to the same resolution. In a previous method [31], the general view set the image resolution to the medium pyramid resolution. However, the general view with the scale sequence (*S*${}^{2}$) module was designed based on $P_{3}$, because small objects are detected in the high-resolution feature map $P_{3}$. Hence, high-resolution feature maps include small object features and localization information. The pyramid feature maps are resized to the resolution of $P_{3}$ by nearest-neighbor upsampling. To construct a general view, the level dimension is added to each resized feature map using the unsqueeze function. Finally, $P_{3}$ and all upsampled feature maps are concatenated.

This general view is fed into the 3D convolution block. The 3D convolution block is composed of 3D convolution, 3D batch normalization, and the leaky ReLU [53] activation function. The 3D convolution is performed along the axis of the pyramid level and extracts a scale sequence feature of the general view. The 3D convolution operation uses a 3 × 3 × 3 (depth, height, width) kernel size with appropriate padding and a stride of 1. As a result, there is no change in resolution size between the input and output. In addition, the number of input channels and output channels is the same. Due to 3D convolution complexity, one convolution block is employed to reduce computation. However, a 3D convolution block can be added in accordance with the purpose.

The output features from the 3D convolution block are computed by average pooling 3D on the level axis only. As a result, the generated general view can compress the features of the level with spatial and channel information preserved. Then, the axis of the level is removed. Finally, the scale sequence features have a width, height, and channel size identical to those of $P_{3}$. For small object detection, both the scale sequence (*S*${}^{2}$) features and $P_{3}$ are combined or used in the detection head together. The new detection head for small objects has the same resolution but twice the channel size, as follows:(5)$$P_{S_{3}^{2}}=CAT\left(P_{3},S^{2}feature\right),$$
where $P_{S_{3}^{2}}$ is the result of concatenation between scale sequence (*S*${}^{2}$) features and $P_{3}$, which is the highest resolution among the pyramid feature maps. As a result, small objects are detected in this new detection head, $P_{S_{3}^{2}}$.

$P_{3}$ is used to extract the proposed scale sequence (*S*${}^{2}$) features for small objects by default. However, the basic resolution size for the scale sequence (*S*${}^{2}$) features does not need to be high. It can be changed to a different resolution depending on the purpose of the application.

The scale sequence (*S*${}^{2}$) module can be applied to both one-stage and two-stage detectors. Figure 3c shows the process of the one-stage detector head, and Figure 3d shows that of the two-stage detector head. In order to modularize the two-stage RoI head effectively, 1 × 1 convolution is added to $P_{S_{3}^{2}}$. As a result, the channel size of $P_{S_{3}^{2}}$ in the two-stage detector is identical to that of $P_{3}$.

### 3.3. Feature-Level Super Resolution Based on Scale Sequence (S${}^{2}$) Features

For small object detection, the super-resolution of images can be an effective approach. Through super resolution, the spatial information of small objects can be enriched so that they are more easily recognizable. However, the super resolution of input images causes a decrease in the inference speed. This would be a serious problem in real-time object detection.

Recently, feature-level super resolution has been studied to reduce computational complexity. This method super-resolves the feature map, not the input image. Unlike super-resolution tasks, object detection predicts bounding boxes and classes based on semantic features from the model. Therefore, it is not necessary to predict the whole image, which would bring a computational burden. Feature-level super resolution is sufficient to create high-quality super-resolved features for performance improvement when using low-resolution images. Noh et al. [49] proposed a super-resolution feature generator based on generative adversarial networks (GANs) [50] for small object detection. Motivated by their approach, a feature-level super-resolution method using scale sequence (*S*${}^{2}$) features is introduced in this study.

Figure 5 shows the proposed feature-level super-resolution method based on scale sequence (*S*${}^{2}$) features for classification tasks. High-resolution (HR)–low-resolution (LR) pairs are used for training. The HR image is easily classified as a whale, but the LR image is difficult to predict. If a model generates an HR image from an LR image, the LR image can be precisely recognized as the desired class.

HR features are extracted from HR images and used as target data for super resolution. The HR encoder uses ResNet-101 as a backbone, and an additional FPN is attached for scale sequence (*S*${}^{2}$) features. The standard feature map of the general view is a $P_{3}$ pyramid feature map, and the size of $P_{3}$ is 8 × 8 × 256 (width, height, channel). HR features are created by concatenating HR $P_{3}$ and HR scale sequence (*S*${}^{2}$) features from the scale sequence module. The size of HR features for training is 8 × 8 × 512.

Before the super-resolution module, the LR encoder is identical to ResNet-101 with the FPN architecture and HR encoder. However, after upsampling the resolution of the HR image via bicubic interpolation, LR images are fed into the LR encoder. As a result, LR $P_{3}$ and LR scale sequence (*S*${}^{2}$) features are extracted for the super-resolution module and concatenated. These features are denoted as LR features.

The super-resolution module is composed of two residual blocks, which were introduced in the enhanced deep super-resolution network (EDSR) [45]. Through the residual blocks, the size of the input feature map remains the same. For feature-level super resolution, pixel shuffling, which is an upsampling step, is removed, because the extracted super-resolved (SR) features do not need to be large. One convolution layer transforms the channel size of the LR feature map from 512 to 1024, and the features are fed into the residual blocks. The residual blocks consist of convolution, ReLU, and convolution layers without a normalization step. Before the residual blocks, one convolution layer transforms the channel size of the LR feature map from 512 to 1024, and the features are used as the residual block input. Then, one convolution layer adjusts the number of channels of the feature map to 512, which is the same as the target number of HR features. The output from the super-resolution module is the final SR feature.

To train the model end-to-end, both classification loss and super-resolution loss are combined as follows:(6)$$L_{total}=\left(1-\alpha \right)*L_{cls}+\alpha *L_{sr},$$
where $L_{total}$ is the total loss, combining the classification loss ($L_{cls}$) with the super-resolution loss ($L_{sr}$); $L_{cls}$ is the cross-entropy loss function; and $L_{sr}$ is the mean squared error (MSE) between HR features and SR features. Since classification is the main task, the model should focus on classification loss above super-resolution loss. α is the weight between both losses. In this paper, α was set to 0.3 for $L_{sr}$ and 0.7 for $L_{cls}$.

In the inference step, LR images from upsampled HR images are fed into the ResNet-101 FPN model. Then, LR features concatenated from LR $P_{3}$ and the *S*${}^{2}$ features of the LR image are extracted. Through the super-resolution module, the SR features are flattened, and, finally, the fully connected layer predicts the classes. As a result, the features of low-resolution images can be super-resolved to HR features that can be recognized easily for classification and detection.

## 4. Experimental Results

### 4.1. Dataset and Evaluation Metrics

All experiments were conducted on the MS COCO 2017 dataset [21]. This dataset is commonly used as a benchmark dataset for object detection and segmentation tasks. It has 80 object categories and consists of a 118 k training set, 5 k validation set, and 20 k test-dev set. We trained the models on the training set without extra data. Evaluations were conducted on the validation set or test-dev set by uploading our model onto the official evaluation server.

All results were evaluated by the average precision (AP) on the MS COCO dataset. We averaged over multiple intersection over union (IoU) values. The primary challenge metric was AP at IoU = 0.50:0.05:0.95, and others were denoted as AP${}_{50}$ at IoU = 0.50 and AP${}_{75}$ at IoU = 0.75. Additionally, we report AP values for different object scales, i.e., split into small AP${}_{S}$, medium AP${}_{M}$, and large AP${}_{L}$ based on the segmentation mask area.

### 4.2. Implementation Details

Experiments were conducted to check the performance improvement when scale sequence (*S*${}^{2}$) features were built on baseline models. For comparison, the training strategy and default configuration were set with each baseline model used in the paper. Pytorch was used for implementation, and pre-trained COCO weights were used as initial weights.

When adding scale sequence (*S*${}^{2}$) features, the neck modules of each baseline model were utilized. For example, the one-stage detector of YOLO used a path aggregation network (PAN) as the neck module. Scale sequence (*S*${}^{2}$) features were implemented based on the PAN. Additionally, the two-stage detectors of detectron2 used a feature pyramid network (FPN). Therefore, the scale sequence (*S*${}^{2}$) features were equipped with FPN. All training procedures were performed by single-scale training without ensemble.

For one-stage detectors, Scaled-YOLOv4 (Scaled-YOLOv4-P5, Scaled-YOLOv4-P6) and YOLOR (YOLOR-P6, YOLOR-W6, YOLOR-D6) were used as baseline models. Hyper-parameters and initial training options followed the settings reported in [54]. Stochastic gradient decent (SGD) was employed as the optimizer, and the learning rate scheduler was OneCycleLR [55], with an initial learning rate of 0.01. One-stage detectors were trained using three Tesla NVIDIA V100 GPUs, and it took about two weeks to train the models. The batch sizes for YOLOv4-P5 and YOLOv4-P5 were 24 and 21, respectively. Additionally, the YOLOR series (YOLOR-P6, YOLOR-W6, YOLOR-D6) was trained with a batch size of 18. The performance of the one-stage detector was evaluated on the MS COCO test-dev dataset.

On the other hand, two-stage detectors were trained using detectron2. Both ResNet-50 and ResNet-101 [56] were selected as the baseline models for Faster R-CNN, Mask R-CNN, and Cascade R-CNN. These three models used an FPN as the neck module. The two-stage detectors were evaluated on the MS COCO validation dataset.

For comparison, the models were re-trained using a batch size of 8. After building the scale sequence (*S*${}^{2}$) features, the two-stage detectors + *S*${}^{2}$ were trained using the same batch size. The training strategy and hyper-parameters were set according to detectron2’s default configuration. The training epochs were 3× scheduled (270 k iterations), and the learning rate decreased by a factor of 0.1 at 210 k and 250 k iterations. The two-stage detectors were trained on four NVIDIA RTX 2080Ti GPUs, and it took about one week to train the models.

### 4.3. Results and Discussion

#### 4.3.1. Overall Performance Analysis

The one-stage detectors’ built-in scale sequence (*S*${}^{2}$) features were evaluated with other YOLO-based models on the MS COCO test-dev dataset. The results are shown in Table 1. All models with scale sequence (*S*${}^{2}$) features consistently improved the performance. YOLOv4-P5 with scale sequence (*S*${}^{2}$) features achieved a 52.3% AP, representing a 0.9% improvement compared to the model without the proposed features. AP${}_{50}$ and AP${}_{75}$ were improved by factors of 0.8% and 1.1%, respectively. Additionally, the AP increased even if the model size was larger. For example, YOLOv4-P6 equipped with scale sequence (*S*${}^{2}$) features demonstrated a 0.5% improvement in all AP metrics, including AP${}_{50}$ and AP${}_{75}$. Furthermore, YOLOR-P6 with scale sequence (*S*${}^{2}$) features achieved 53.1% AP, representing a 0.5% improvement compared to the model without the proposed features. Both AP${}_{50}$ and AP${}_{75}$ were improved by a factor of 0.5%. The state-of-the-art architecture YOLOR-D6 achieved 55.4% AP using the proposed scale sequence (*S*${}^{2}$) feature (bold face).

Furthermore, Table 2 shows a comparison of our scale sequence feature with two-stage object detector baselines. These experiments were only conducted on the MS COCO validation set. Unlike the one-stage detectors, the scale sequence (*S*${}^{2}$) module for two-stage detectors added 1 × 1 convolution to modularize the ROI head effectively. Faster R-CNN, Mask R-CNN, and Cascade R-CNN were evaluated with the proposed method.

As shown in Table 2, Faster R-CNN with the scale sequence (*S*${}^{2}$) features achieved a 39.1% AP, which was 1.2% higher than that of Faster R-CNN without the proposed feature. Additionally, AP${}_{50}$ and AP${}_{75}$ were improved by 1.4% and 1.2%, respectively.

For Mask R-CNN and Cascade R-CNN, the proposed scale sequence (*S*${}^{2}$) feature improved the AP as well as AP${}_{mask}$. Mask R-CNN with scale sequence features was improved by a factor of 1.3% in terms of AP and 1.1% in terms of AP${}_{mask}$. Additionally, Cascade R-CNN achieved a 43.2% AP and 37.5% AP${}_{mask}$, which were 1.3% and 1.0% higher than the model without our feature, respectively. The proposed scheme also significantly improved the AP${}_{50}$ and AP${}_{75}$.

#### 4.3.2. Analysis of the Performance in Terms of Object Scale

AP improvement for different object scales: small, medium, and large scales were analyzed. All YOLO-based one-stage detectors with scale sequence (*S*${}^{2}$) features increased the AP for all scales. In particular, AP${}_{S}$ increased more than the other scales. This was because the scale sequence feature was designed based on a high-resolution pyramid feature map for small objects. YOLOv4-P5, which used the smallest resolution as the input image, achieved 1.1% and 1.3% AP improvements for AP${}_{L}$ and AP${}_{S}$, respectively.

On the other hand, YOLOv4-P6 was more complex and used larger input resolutions than YOLOv4-P5. When the scale sequence (*S*${}^{2}$) feature was added to YOLOv4-P6, it showed the highest improvement of 1.1% in AP${}_{S}$, followed by 0.3% in AP${}_{M}$ and 0.4% in AP${}_{L}$.

Additionally, YOLOR-D6 with the scale sequence (*S*${}^{2}$) features showed improvements in the primary AP. We observed an increased AP${}_{S}$ and AP${}_{L}$, but AP${}_{M}$ decreased slightly. For the YOLOR series architecture, the pyramid feature map for medium objects had the smallest number of channels when *S*${}^{2}$ was concatenated to a pyramid feature map for small objects. This caused a slight drop in AP${}_{M}$, but the overall AP was improved.

Furthermore, two-stage detectors equipped with the scale sequence (*S*${}^{2}$) feature could improve the AP for all scales consistently. However, the highest improvement in AP among the scales differed, because the proposed scale sequence (*S*${}^{2}$) features were added to *P*${}_{3}$ through 1 × 1 convolution to adjust the feature channel in the two-stage scale sequence (*S*${}^{2}$) module. This caused a lack of information in *P*${}_{3}$ compared to the one-stage detectors.

Figure 6 visualizes the MS COCO bounding box results for small objects (kite and snowboard) with and without scale sequence (*S*${}^{2}$) features. On the left are the outputs from YOLOv4-P6, and on the right are the outputs from YOLOv4-P6 + *S*${}^{2}$ (proposed) for the MS COCO dataset. With the scale sequence (*S*${}^{2}$) features, small objects were more efficiently detected.

### 4.4. Ablation Study

#### 4.4.1. Ablation Study for Different Pyramid Level Positions

Ablation experiments were conducted to explore the number of scale sequence (*S*${}^{2}$) features and different pyramid level positions. Table 3 shows the results. Under the default settings, the proposed scale sequence (*S*${}^{2}$) features were generated based on *P*${}_{3}$ and had the same resolution of *P*${}_{3}$. We resized these features to other pyramid resolutions, *P*${}_{4}$ and *P*${}_{5}$. Finally, the scale sequence (*S*${}^{2}$) features were concatenated to *P*${}_{4}$, *P*${}_{5}$, or both.

As a result, models with the scale sequence (*S*${}^{2}$) features showed improved performance compared to those without the proposed features. When concatenating the scale sequence (*S*${}^{2}$) features to only *P*${}_{3}$, we achieved the best performance. This improved the AP${}_{S}$ as well as the AP for other scales with low complexity.

#### 4.4.2. Ablation Study on Neck Model

To analyze the effect of different necks, the neck module of Scaled-YOLOv4 was changed from PAN to FPN. Table 4 shows the results of this ablation study. Scale sequence (*S*${}^{2}$) features were extracted from FPN instead of PAN. As a result, the model with scale sequence (*S*${}^{2}$) features generated from PAN had better performance.

Path aggregation networks (PANs) have bi-directional aggregation paths from FPNs. These connect feature pyramids through top-down and bottom-up pathways and make all feature pyramids reflect each other; however, they only have different feature map sizes on the level axis. These feature pyramids resemble scale space. Therefore, scale-invariant features can easily be extracted from scale space.

### 4.5. Runtime Analysis

As shown in Table 5, the number of model parameters and the inference speed were analyzed when adding the proposed scale sequence (*S*${}^{2}$) features (blod faces). Models with the scale sequence (*S*${}^{2}$) features showed parameter increased of approximately 2 M. Additionally, the runtime with a batch size of 8 on a NVIDIA Tesla V100 was tested. The speed of YOLOv4-P5 with the scale sequence (*S*${}^{2}$) features was increased by 1.7 ms using an 896 × 896 input image. When using a 1280 × 1280 image as input, the speed was increased by 4.7 ms for YOLOv4-P6 with the scale sequence (*S*${}^{2}$) features. We could see by the runtime that the proposed *S*${}^{2}$ features did not introduce a high complexity.

### 4.6. Feature-Level Super-Resolution Results

This section presents the results of the feature-level super-resolution approach using scale sequence (*S*${}^{2}$) features. The purpose of the experiment was to demonstrate whether super-resolved features could improve the classification performance compared to training using only low-resolution images.

#### 4.6.1. Implementation Details

Experiments were conducted using classification tasks on the CIFAR-100 [60] dataset. The HR images were the original CIFAR-100 images that had a resolution of 32 × 32. On the other hand, the LR images were created by downsampling ×2. Therefore, the LR images had a resolution of 16 × 16. ResNet-101 was used as a baseline model, and the additional FPN architecture was equipped with the baseline to extract scale sequence (*S*${}^{2}$) features. For a fair comparison, all models were trained using 100 epochs and a batch size of 512. Models were trained with an SGD optimizer by setting the learning rate to 0.1, using a Tesla NVIDIA V100 GPU. It took about five hours to train the models.

#### 4.6.2. Classification Results

Table 6 shows the classification accuracy results of ResNet-101 as the baseline model, trained on LR or HR images. Additionally, it shows how much the performance improved with the addition of scale sequence (*S*${}^{2}$) features and the feature-level super-resolution approach. The performance of ResNet-101 trained on HR images was 53.6%. ResNet-101 with scale sequence features trained on HR images achieved a 57.2% classification accuracy, which was 3.6% higher than without the scale sequence (*S*${}^{2}$) features.

However, training on LR images achieved a poorer performance than training on HR images. For example, the classification accuracy of ResNet-101 trained on LR images was 47.1%, which was 6.5% lower than that of ResNet-101 trained on HR images. Furthermore, the experimental results demonstrated that the scale sequence (*S*${}^{2}$) features could improve the classification accuracy of small-image data such as CIFAR-100 data (32 × 32) (even if the CIFAR-100 resolution was downsampled to 16 × 16). For example, when training on LR images, ResNet-101 with *S*${}^{2}$ features achieved a 49.1% classification accuracy, which was 2% higher than without scale sequence (*S*${}^{2}$) features. Nevertheless, it demonstrated a poorer performance than models trained on HR images. The accuracy of ResNet-101 with scale sequence (*S*${}^{2}$) features trained on LR images was 8.1% and 4.5% lower than that of ResNet-101 trained on HR images with and without *S*${}^{2}$ features, respectively.

However, the proposed feature-level super-resolution method using scale sequence (*S*${}^{2}$) features could improve the performance of models trained on LR images to approach that of models trained on HR images. By training using the feature-level super-resolution approach, ResNet-101 with scale sequence (*S*${}^{2}$) features trained on LR images achieved a 55.2% accuracy, which was 6.1% higher than without the super-resolution method. Furthermore, this result was 2% lower than when an HR encoder was used to generate target HR features but 1.6% higher than when ResNet-101 was trained on HR images. Thus, the results showed that the scale sequence (*S*${}^{2}$) feature was effective for feature-level super resolution.

## 5. Conclusions

In this paper, we proposed a new scale sequence (*S*${}^{2}$) feature for improved object detection. It is extracted from the neck module of object detection models such as the FPN. The proposed feature could enrich FPN features by reflecting a sequence of convolution that has not been considered before. In particular, the proposed feature was designed based on high-resolution pyramid feature maps for improving small object detection. In addition, scale sequence (*S*${}^{2}$) features could be simply extended to most object detection models with FPNs. In addition, a feature-level super-resolution approach using scale sequence (*S*${}^{2}$) features was proposed. Both LR and HR encoders were based on FPNs with a scale sequence module, and additional residual blocks generated super-resolved features for prediction. Feature-level super-resolution involves less computational burden, because it is not necessary to recreate the original image resolution.

In the experiments, we achieved a noticeable improvement in AP for small as well as other scales. Additionally, we demonstrated that both one-stage and two-stage detectors with scale sequence (*S*${}^{2}$) features increased the AP on the MS COCO dataset. Based on the scale sequence (*S*${}^{2}$) features, we achieved up to 0.9% and 1.4% AP improvements for YOLOv4-P5 and Mask R-CNN, respectively. Furthermore, YOLOv4-P6 and Faster R-CNN with our scale sequence (*S*${}^{2}$) features achieved AP${}_{S}$ improvements of up to 1.1% and 1.2%, respectively.

For the super-resolution approach, an experiment was conducted for classification on the CIFAR-100 dataset, which was composed of small-resolution images. The proposed feature-level super-resolution approach using scale sequence (*S*${}^{2}$) features could improve the performance of models trained on LR images to approach that of models trained on HR images. ResNet-101 with scale sequence (*S*${}^{2}$) features trained on LR images achieved a 55.2% classification accuracy, which was 6.1% higher than that without the super-resolution method. Additionally, the designed scheme showed a 1.6% higher accuracy than ResNet-101 trained on HR images.

In future work, we need to conduct experiments on feature-level super resolution with scale sequence features for object detection model. If small object features are super-resolved, the AP for small objects could be improved. Furthermore, research on shortening the inference and training times is needed. Additionally, model robustness against noise must be assessed.

## Figures and Tables

**Figure 1 sensors-23-04432-f001:**
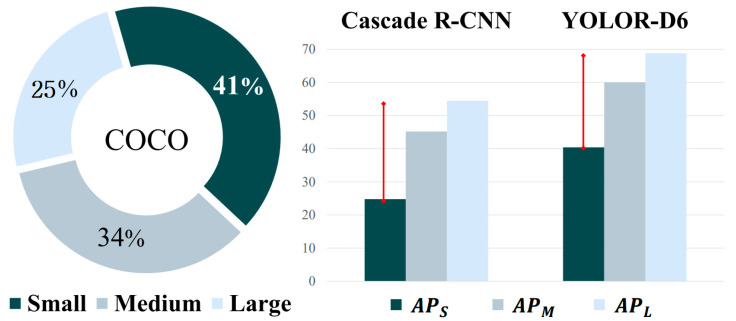
The proportions of different object scales in the MS COCO dataset and the gap in AP between small-, medium-, and large-scale objects in the MS COCO validation dataset.

**Figure 2 sensors-23-04432-f002:**
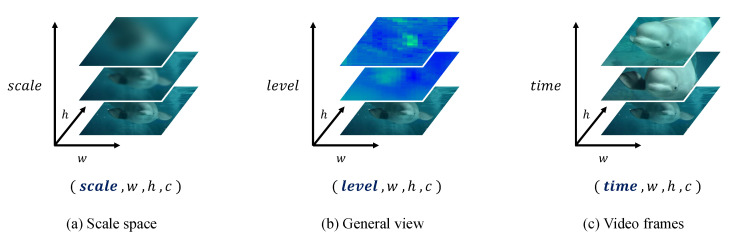
Comparison of image sequences on different axes. The source of the images was the YouTube-8M dataset [51]. (**a**) Scale space using a Gaussian filter on the scale axis. (**b**) General view concatenated with identical resolution features on the level axis [31]. (**c**) Video frames on the time axis.

**Figure 3 sensors-23-04432-f003:**
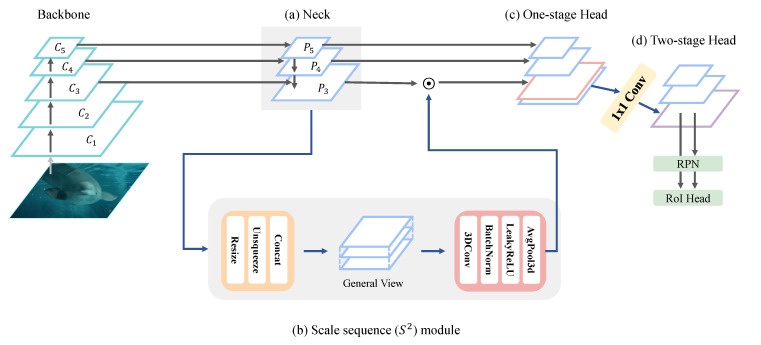
Scale sequence module framework: (**a**) neck module of FPN for feautre fusion, (**b**) process of proposed scale sequence module using 3D convolution, (**c**) one-stage detector head, (**d**) two-stage detector head.

**Figure 4 sensors-23-04432-f004:**
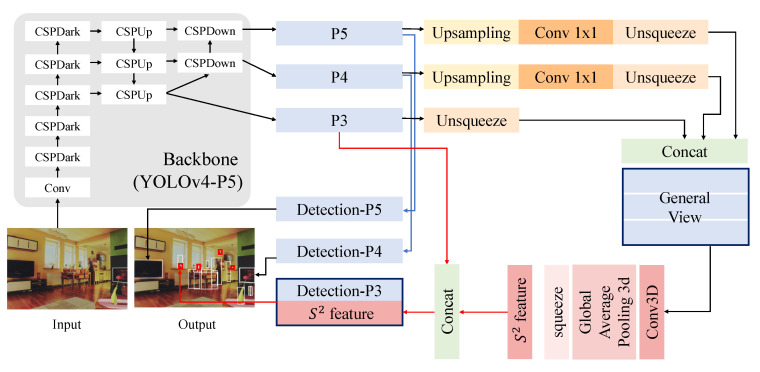
The architecture of the scale sequence (*S*${}^{2}$) module in detail.

**Figure 5 sensors-23-04432-f005:**
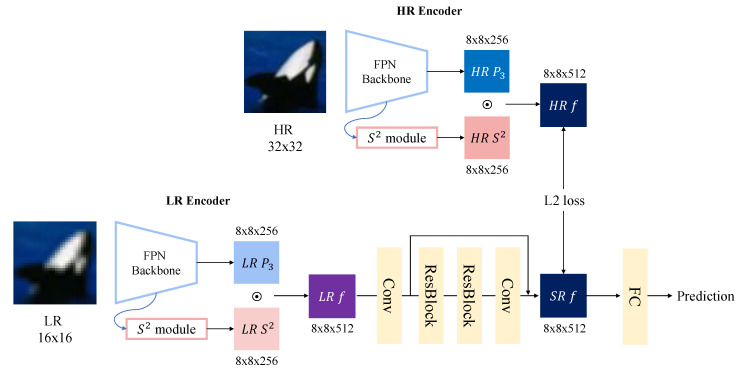
The architecture of the feature-level super-resolution model with scale sequence features.

**Figure 6 sensors-23-04432-f006:**
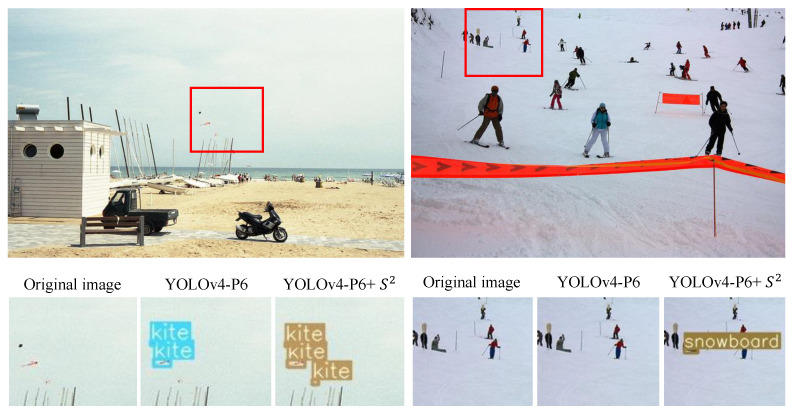
Comparison of bounding box results for small objects between YOLOv4-P6 and the proposed scheme on MS COCO dataset.

**Table 1 sensors-23-04432-t001:** Comparison of the one-stage detectors with scale sequence (*S*${}^{2}$) features and baseline models evaluated on COCO test-dev dataset.

Model	Backbone	Size	AP (%)	AP${}_{50}$ (%)	AP${}_{75}$ (%)	AP${}_{S}$ (%)	AP${}_{M}$ (%)	AP${}_{L}$ (%)
DyHead	ResNext-64× 4d-101	-	47.7	65.7	51.9	31.5	51.7	60.7
PyCenterNet [57]	Swin-L	-	53.2	71.4	57.4	33.2	56.2	68.7
DetectoRS [58]	ResNeXt-101-32×4d	-	53.3	71.6	58.5	33.9	56.5	66.9
QueryInst [59]	Swin-L	-	56.1	74.2	53.8	31.5	51.8	63.2
YOLOv4-P5	CSP-P5	896	51.4	69.9	56.3	33.1	55.4	62.4
YOLOv4-P6	CSP-P6	1280	54.3	72.3	59.5	36.6	58.2	65.5
YOLOR-P6	CSPdarknet53	1280	52.6	70.6	57.6	34.7	56.6	64.2
YOLOR-W6	CSPdarknet53	1280	54.1	72.0	59.2	36.3	57.9	66.1
YOLOR-D6	CSPdarknet53	1280	55.3	73.3	60.6	38.0	59.2	67.1
**YOLOv4-P5 + *S*${}^{2}$**	CSP-P5	896	52.3[+0.9]	70.7**[+0.8]**	57.4**[+1.1]**	34.2**[+1.1]**	56.2**[+0.8]**	63.7**[+1.3]**
**YOLOv4-P6 + *S*${}^{2}$**	CSP-P6	1280	54.8**[+0.5]**	72.8**[+0.5]**	60.0**[+0.5]**	37.7**[+1.1]**	58.5**[+0.3]**	65.9**[+0.4]**
**YOLOR-P6 + *S*${}^{2}$**	CSPdarknet53	1280	53.1**[+0.5]**	71.2**[+0.6]**	58.2**[+0.6]**	35.6**[+0.9]**	56.5**[–0.1]**	64.7**[+0.5]**
**YOLOR-W6 + *S*${}^{2}$**	CSPdarknet53	1280	54.2**[+0.1]**	72.3**[+0.3]**	59.4**[+0.2]**	36.7**[+0.4]**	57.8**[–0.1]**	66.2**[+0.1]**
**YOLOR-D6 + *S*${}^{2}$**	CSPdarknet53	1280	55.4**[+0.1]**	73.5**[+0.2]**	60.0**[+0.0]**	38.1**[+0.1]**	58.9**[–0.3]**	67.2**[+0.1]**

**Table 2 sensors-23-04432-t002:** Comparison of the two-stage detectors with scale sequence (*S*${}^{2}$) features and baseline models evaluated on COCO validation dataset.

Model	Backbone	AP (%)	AP${}_{mask}$ (%)	AP${}_{50}$ (%)	AP${}_{75}$ (%)	AP${}_{S}$ (%)	AP${}_{M}$ (%)	AP${}_{L}$ (%)
Faster R-CNN	ResNet-50	37.9	-	58.1	41.3	22.0	40.9	49.1
Faster R-CNN	ResNet-101	39.7	-	60.1	43.3	23.5	43.4	51.4
Mask R-CNN	ResNet-50	38.5	35.1	58.7	42.0	22.4	41.4	49.9
Mask R-CNN	ResNet-101	40.6	36.9	61.1	44.0	24.6	44.0	52.7
Cascade R-CNN	ResNet-50	41.9	36.5	59.6	45.4	24.8	45.2	54.4
**Faster R-CNN + *S*${}^{2}$**	ResNet-50	39.1**[+1.2]**	-	59.6**[+1.4]**	42.5**[+1.2]**	23.3**[+1.2]**	43.0**[+2.0]**	50.9**[+1.7]**
**Faster R-CNN + *S*${}^{2}$**	ResNet-101	41.3**[+1.6]**	-	61.6**[+1.5]**	44.8**[+1.5]**	25.2**[+1.2]**	44.9**[+1.5]**	53.5**[+2.1]**
**Mask R-CNN + *S*${}^{2}$**	ResNet-50	39.8**[+1.3]**	36.2[+1.1]	60.0**[+1.3]**	43.6**[+1.6]**	23.5**[+1.1]**	43.0**[+1.6]**	51.1**[+1.2]**
**Mask R-CNN + *S*${}^{2}$**	ResNet-101	42.0**[+1.4]**	37.9[+1.0]	62.1**[+1.0]**	46.0**[+2.0]**	25.7**[+1.1]**	45.4**[+1.4]**	54.0**[+1.3]**
**Cascade R-CNN + *S*${}^{2}$**	ResNet-50	43.2**[+1.3]**	37.5[+1.0]	60.8**[+1.3]**	47.1**[+1.7]**	25.8**[+1.0]**	46.6**[+1.4]**	56.4**[+2.0]**

**Table 3 sensors-23-04432-t003:** Ablation study on different pyramid level positions for concatenating the scale sequence (*S*${}^{2}$) features.

Level	AP (%)	AP${}_{S}$ (%)	AP${}_{M}$ (%)	AP${}_{L}$ (%)
YOLOv4-P5	51.4	33.1	55.4	62.4
*P*_3_ + *S*${}^{2}$	52.3	34.2[+1.1]	56.2[+0.8]	63.7[+1.3]
*P*_3_, *P*_4_ + *S*${}^{2}$	52.2	34.2[+1.1]	56.4[+1.0]	63.3[+0.9]
*P*_3_, *P*_4_, *P*_5_ + *S*${}^{2}$	52.2	33.7[+0.6]	56.5[+1.1]	63.2[+0.8]

**Table 4 sensors-23-04432-t004:** Ablation study on different neck modules (FPNs).

Model	AP (%)	AP${}_{50}$ (%)	AP${}_{75}$ (%)
YOLOv4-P5			
w PAN	51.4	69.9	56.3
w FPN + *S*${}^{2}$	47.5	67.6	51.7
w PAN + *S*${}^{2}$	52.3	70.7	57.4

**Table 5 sensors-23-04432-t005:** Comparison of runtime analysis.

Model	Size	Param. (M)	AP (%)	AP${}_{50}$ (%)	Speed (ms)
YOLOv4-P5	896	71	51.4	69.9	11.8
YOLOv4-P6	1280	128	54.3	72.3	23.7
YOLOR-P6	1280	37	52.6	70.6	12.4
YOLOR-W6	1280	80	54.1	72.0	14.0
**YOLOv4-P5 + *S*${}^{2}$**	896	73	52.3	70.7	13.5**[+1.7]**
**YOLOv4-P6 + *S*${}^{2}$**	1280	130	54.8	72.8	28.4**[+4.7]**
**YOLOR-P6 + *S*${}^{2}$**	1280	40	53.1	71.2	16.6**[+4.2]**
**YOLOR-W6 + *S*${}^{2}$**	1280	82	54.2	72.3	18.3**[+4.3]**

**Table 6 sensors-23-04432-t006:** Performance comparison of super-resolution results and baseline model on CIFAR-100 dataset.

Image Size	Model	SR	Accuracy (%)	Epochs
HR (32 × 32)	ResNet-101 + ***S*${}^{2}$**	×	57.2	100
HR (32 × 32)	ResNet-101	×	53.6	100
LR (16 × 16)	ResNet-101 + ***S*${}^{2}$**		**55.2**	100
LR (16 × 16)	ResNet-101 + ***S*${}^{2}$**	×	49.1	100
LR (16 × 16)	ResNet-101	×	47.1	100

## Data Availability

https://github.com/smu-ivpl/ssFPN.git (1 February 2023).

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
