# Peer review of "ssFPN: Scale Sequence (S2) Feature-Based Feature Pyramid Network for Object Detection"

_sensors, 2023, doi:10.3390/s23094432_

Round 1

Reviewer 1 Report

This manuscript proposed a novel deep learning-based approach for object detection, where a new FPN model was employed for the task of interest. In the proposed model, the scale sequence feature is extracted from FPN to strengthen information on small objects based on scale-space theory.The performance of the proposed scheme was validated using MS COCO dataset, with satisfactory results. Overall, the topic of this research is interesting, and the manuscript was well organised and written. The detailed comments are given as follows.

1.       The main innovation and contributions of this research should be well clarified in both abstract and introduction.

2.       Broaden and update literature review on deep learning and its application in data processing. E.g. Vision-based concrete crack detection using a hybrid framework considering noise effect; Torsional capacity evaluation of RC beams using an improved bird swarm algorithm optimised 2D convolutional neural network

3.       The performance of deep learning models is mainly affected by setting of hyperparameters. How did the author select optimal network hyperparameters to achieve the optimal object detection

4.       Training time should be considered as a metric for performance evaluation.

5.       How about the robustness of the proposed method against noise effect?

6.       More future research should be included in conclusion part.

Author Response

Dear Reviewer, 

Thank you for your valuable comments and suggestion.  I have attached my reply letter as file. Could you refer to the attachment?

Reviewer 2 Report

Summary:

In this work, the authors propose a new method, Scale Sequence(S²), to explore the scale-invariance of FPN. S²uses the horizontal axis of FPN as the time axis of a sequence and extracts spatiotemporal features using 3D convolution, improving the detection of small objects. Furthermore, based on S²features, a super-resolution method is proposed to improve the shortcomings of existing feature-level super-resolution methods and enhance the running efficiency. The paper achieves good results on the COCO dataset and conducts a large number of experiments to verify the feasibility of the work.

Advantages:

1The paper proposes a novel S²feature to enhance the detection of small objects, which is a novel idea.

2The paper conducts sufficient experiments on one-stage and two-stage models, and the experiments on the COCO dataset verify the superiority of the proposed method.

3In the third chapter, the paper systematically sorts out from theory to its own model, with clear organization, which helps readers understand.

4The authors provide source code for reference, which helps readers better understand the proposed method.

Disadvantages:

1Related Work introduces the relevant content of SEPC, which needs an index.

2The contribution part in the Introduction does not mention the improvement of feature-level super-resolution later.

3The second point in the proposed method is Feature-level Super Resolution based on Scale Sequence (S²) Feature, which proposes its own method to improve feature-level super-resolution, but there is no introduction to feature-level super-resolution in Related Work.

4The title content of Table 4 in the ablation experiment is written incorrectly and should be a comparison of different FPNs. Moreover, I think experimental data for w FPN can be added to Table 4.

5The paper mentions that S²features are established only on high resolution (P3) for small objects, and experiments are conducted on the establishment position of S²features. Why not conduct multiple sets of S² feature experiments, such as (P3+S², P4+S²) and other comparisons?

6,some important paper should be cited:a, Siamese implicit region proposal network with compound attention for visual tracking [J]. IEEE Transactions on Image Processing, 2022, 31: 1882-1894.; b , Online multiple object tracking using joint detection and embedding network [J]. Pattern Recognition, 2022, 130: 108793.

Author Response

(The authors gave the same response as above.)

Round 2

Reviewer 2 Report

All my comments have modified. I think it can be published.